# MicroRNA205: A Key Regulator of Cardiomyocyte Transition from Proliferative to Hypertrophic Growth in the Neonatal Heart

**DOI:** 10.3390/ijms25042206

**Published:** 2024-02-12

**Authors:** Jonathan J. Weldrick, Rui Yi, Lynn A. Megeney, Patrick G. Burgon

**Affiliations:** 1Department of Cellular and Molecular Medicine, Faculty of Medicine, University of Ottawa, Ottawa, ON K1N 6N5, Canada; jweld090@uottawa.ca (J.J.W.); lmegeney@ohri.ca (L.A.M.); 2Department of Pathology, Northwestern University Feinberg School of Medicine, Chicago, IL 60611, USA; yir@northwestern.edu; 3Department of Dermatology, Northwestern University Feinberg School of Medicine, Chicago, IL 60611, USA; 4Sprott Centre for Stem Cell Research, Ottawa Hospital Research Institute, Ottawa Hospital, Ottawa, ON K1Y 4E9, Canada; 5Department of Medicine, Faculty of Medicine, University of Ottawa, Ottawa, ON K1N 6N5, Canada; 6Department of Chemistry and Earth Sciences, College of Arts and Sciences, Qatar University, Doha P.O. Box 2713, Qatar

**Keywords:** postnatal heart development, Hippo pathway, hypertrophy

## Abstract

The mammalian myocardium grows rapidly during early development due to cardiomyocyte proliferation, which later transitions to cell hypertrophy to sustain the heart’s postnatal growth. Although this cell transition in the postnatal heart is consistently preserved in mammalian biology, little is known about the regulatory mechanisms that link proliferation suppression with hypertrophy induction. We reasoned that the production of a micro-RNA(s) could serve as a key bridge to permit changes in gene expression that control the changed cell fate of postnatal cardiomyocytes. We used sequential expression analysis to identify miR205 as a micro-RNA that was uniquely expressed at the cessation of cardiomyocyte growth. Cardiomyocyte-specific miR205 deletion animals showed a 35% increase in heart mass by 3 months of age, with commensurate changes in cell cycle and Hippo pathway activity, confirming miR205’s potential role in controlling cardiomyocyte proliferation. In contrast, overexpression of miR205 in newborn hearts had little effect on heart size or function, indicating a complex, probably redundant regulatory system. These findings highlight miR205’s role in controlling the shift from cardiomyocyte proliferation to hypertrophic development in the postnatal period.

## 1. Introduction

During fetal and early perinatal development, the mammalian myocardium undergoes a period of hyperplastic growth, which results in an exponential increase in the number of cardiomyocytes [1,2]. Soon after birth, cardiomyocytes permanently exit the cell cycle due to loss of cytokinesis, resulting in the formation of polyploid cardiomyocytes. As a consequence, the postnatal increase in myocardial mass is accomplished exclusively through the hypertrophy of the cardiomyocyte population [2,3,4,5]. This early postnatal loss in proliferation competence results in an adult mammalian heart with limited repair capacity, even though the fetal heart has an intrinsic repair competence [6]. Nevertheless, very little is known about the framework that controls the transition from the fetal heart gene program to the adult heart gene program [7]. For example, 1–3-day-old perinatal murine hearts injured via surgical resection completely regenerate with no scar or functional deficit observed three weeks later. However, when a 7-day-old neonatal heart was similarly injured, the heart had a greatly diminished capacity for repair, suggesting a fundamental change in the somewhat older cell population [6].

Despite numerous descriptive studies characterizing the limited ability of adult mammalian cardiomyocytes to divide in response to various stimuli, no definitive mechanisms have been identified to explain why the majority of adult mammalian cardiomyocytes are unable to re-enter the cell cycle when stimulated appropriately [8,9,10,11]. Interestingly, the neonatal heart’s regenerative potential is closely associated with an increased ploidy, that may in turn be reflected by binucleation or by endoreduplication of the cardiomyocyte genome without nuclear division (karyokinesis) [12,13].

These observations suggest that a well-orchestrated gene expression and chromatin remodeling event must exist that ‘locks’ cardiomyocytes into a non-proliferative state. Specifically, the binucleated mammalian adult cardiomyocyte is unable to undergo cytokinesis despite still being able to engage in endoreduplication [14]. Further, we surmise that these genomic remodeling activities are initiated and established during the perinatal heart’s transition from hyperplastic to hypertrophic growth. MicroRNAs may offer an idealized mechanism to manage such a terminal transition, as the expression of these molecules are rapidly induced within a target cell, while retaining the capacity to target and repress one or multiple regulatory transcripts.

MicroRNAs (miRNA) play an essential role in genome regulation, downregulating gene expression in a variety of manners, including translational repression, mRNA cleavage, and deadenylation. A single miRNA is able to modulate as many as 200 different genes. miRNAs are expressed in the heart and have been demonstrated to regulate cardiac development and growth [15,16,17]. Of note, the cardiac-specific ablation of Dicer, an enzyme that facilitates the activation of the RNA-induced silencing complex (RISC, essential for micro-RNA interference) results in neonatal lethality [18]. Members of the miR-15 family have been reported to be upregulated in the infarct region after ischemic injury [19] and that therapeutic targeting of miR-15 in mice reduces infarct size and cardiac remodeling and enhances cardiac function in response to myocardial infarction [19]. Despite numerous studies that investigated the cardiac role of miRNAs, miRNAs’ involvement in the perinatal transition to adulthood remains undefined. We surmise that one or more miRNAs act as primary genetic drivers which irreversibly commit myocardial cells to the adult genotype.

Here, we demonstrate that one micro-RNA, miR205, displays a unique expression pattern consistent with managing cardiac gene expression during the transition from a fetal to the adult heart. Loss of function of miR205 results in uncontrolled cardiomyocyte growth with attendant pathology, suggesting that this micro-RNA serves as a genomic conduit to establish the adaptive hypertrophy phenotype of adult heart muscle cells.

## 2. Results

To investigate the role micro-RNAs may play in the transition from hyperplastic to hypertrophic growth, we first defined the precise timing of the heart’s cell cycle withdrawal for the cardiomyocyte population. The cardiomyocyte withdrawal from cell cycle was based on the appearance of binucleated cardiomyocytes, which form the vast majority of the murine adult myocardium and are no longer cytokinesis-competent [2,20]. Prior work in this area has been hampered by the lack of an amenable method for isolation of mouse cardiomyocytes beyond 1 day post-birth. Therefore, to address the relevant biology, we concurrently developed a reliable cardiomyocyte isolation method that allowed us to efficiently isolate cells from neonate to adult hearts [21]. Based on the cardiomyocyte binucleation data [2,20] and the pattern of the temporal expression profile of key cell cycle checkpoint markers, phospho-Rb (G1/S) and phosphoCdc2 (G2/M) [20], we established the appropriate timeframe to monitor micro-RNA expression levels in the neonatal heart.

Cardiac tissue was collected at the following time points: embryonic day 19 (E19); and 1, 3, 5, 7, 10, and 35 days post-birth (Figure 1A). Expression profiling of a miRNA library revealed that numerous micro-RNAs, many of which had proven roles in heart development and disease expression levels, had expression levels which were altered during perinatal heart development. We assessed the temporal expression profile of all micro-RNAs that were significantly upregulated in the first-week post-birth (Figure 1B), and identified one micro-RNA that was uniquely matched to the cell cycle/cytokinesis transition point in the cardiomyocyte population namely, micro-RNA205 (miR205) (Figure 1B).

miR205 has been studied extensively in cancer and stem cell models. It has been reported to be enriched in skin stem cells and is highly associated with undifferentiated cell populations (14), but is not required for the generation or maintenance of stem cells [22]. To date, there have been reports on the role of miR205 in the heart [23,24,25], but no reports describing miR205 in the neonatal heart, despite extensive investigations into the mapping and characterization of micro-RNAs that modulate heart development and adult cardiac disease. miR205 is the most robustly upregulated micro-RNA during the first five days post-birth, which we confirmed by RT-qPCR (Figure 1C). In addition, miR205 displayed a singularly unique and transient ‘burst’ of expression that peaked 3 to 5 days post-birth and returned to pre-birth levels by 7 days post-birth (Figure 1C). No other micro-RNA matched this expression profile. In situ hybridization analysis of miR205 spatial expression revealed that miR205 is most abundantly expressed in the nuclei of the epicardium and subepicardial myocardium of 5-day-old neonatal hearts (Figure 1D).

To determine if miR205 was essential for heart growth and adaptation we established cardiac specific miR205 mouse models. We generated a cardiomyocyte-restricted deletion of miR205 (miR205cKO) through the crossing of the miR205^flox/flox^ [22] to miR205^flox/+^; αMHC–Cre^+/−^ mice were maintained on a C57BL/6 background. These mice were born with an expected Mendelian ratio and were indistinguishable from their non-αMHC-Cre littermates. We then crossed this compound heterozygote mouse with a homozygote floxed miR205 (miR205^fl/fl^) mouse to generate mice that had cardiomyocyte-specific ablation of the miR205 in their neonatal/postnatal hearts. At this time, it should be noted that several published reports have demonstrated that overexpression of Cre recombinase in mouse cardiomyocytes is cytotoxic and leads to dilated cardiomyopathy and premature death [26,27,28]. However, a recent study concluded that αMHC-Cre animal models are effective in young mice but are not suitable for older animals [29]. For these reasons, we focused our analysis primarily on the first two weeks post-birth.

Baseline echocardiographic analysis of these mice at 12 weeks of age revealed an approximately 30–50% increase in the LV mass with no functional deficit (Table 1). A noticeable and significant increase in the heart mass (~+35%) of the miR205-deficient mice was observed at 6 months of age (Figure 2A). To determine if the observed increase in heart mass of miR205-deficient hearts was the result of an aberrant cell cycle, we performed Western blot analysis of cell cycle checkpoint phospho-CDK1 (Figure 2B), immunohistochemical analysis of Ki67 and phospho-Histone 3 (Figure 2C). Elevated levels of phospho-CDK1 were observed beyond neonate day 5 (Figure 2B), which was corroborated by an observed increased signal for both Ki67 and phospho-Histone 3 (Figure 2C). These observations were further corroborated by increased expression of proliferative markers (Ki67 and pH3) determined by immunohistochemistry at 14 days post-birth (Figure 2D,E). Furthermore, we also observed a significant increase in cell number in the 14-day-old miR205null hearts (Figure 2G). While these findings are compelling in their support of miR205 regulating cell proliferation in the heart, it should be noted these observations are correlative.

We next determined if the Hippo pathway was activated in the miR205-deficient neonatal hearts. The Hippo pathway is a conserved signaling cascade that is essential for the proper regulation of organ growth by regulating cell number. In the adult heart, activation of upstream kinase components of the Hippo pathway prevents adult cardiomyocyte proliferation and regeneration [30], whereas direct activation of the Hippo pathway target, the transcription factor YAP1, results in cardiomyocyte proliferation [31,32,33,34,35,36]. Western blot analysis of key components of the Hippo pathway revealed an increase in Hippo pathway activity, with YAP1 being elevated throughout the neonatal period in the miR205-deficient hearts (Figure 1D). Subsequently, we examined whether the enlarged miR205 null hearts originated (in part) from an increased number of cardiomyocytes, owing to elevated cell cycle and Hippo pathway activity. Histological examination of 14-day-old hearts revealed a significant increase in the density of cardiomyocytes (Figure 2G), matching the elevated markers of cell cycle progression (Figure 2F).

Next, we sought to address the impact of overexpressing miR205 in the adult heart. We generated mice with doxycycline-inducible CM-restricted overexpression of miR205 (miR205^iOE^) by crossing the TetRE–miR205 mouse line (a B6SJLF1/J transgenic line in which miR205 is placed under the control of a minimal CMV promoter and the tetracycline-responsive element) with αMHC–tTA mice [37]. As such, this model expressed tetracycline-responsive transcriptional activator (tTA) under the control of the human alpha myosin heavy chain promoter (αMHC) promoter (Figure 3A). Two days prior to expected birth, doxycycline (DOX) was administered to the pregnant mice in their drinking water (2 mg/mL) to induce transgene expression, and we observed an ~7-fold induction of miR205 in the neonatal hearts within 1–2 days of DOX-mediated induction (Appendix A).

Unexpectedly, we observed a limited increase in CDK1 and pRb activity (Figure 3B), which was mirrored by activation of the Hippo pathway (Figure 3C). However, this limited increase in both the cell cycle and Hippo pathway was not associated with an increased number of cardiomyocytes (Figure 4B), suggesting that ectopic overexpression of miR205 may have a finite impact on the neonatal heart. We suggest that this limited impact may arise from the probability that endogenous miR205 may have already saturated all miR205 targets, especially since micro-RNA bound to its target mRNA results in their degradation.

## 3. Discussion

In the present study, we demonstrated, for the first time, that miR205 is transiently expressed in the mouse heart during the neonatal period when the mammalian cardiomyocyte population begins to exit the cell cycle in the absence of cytokinesis and causes a subsequent increase in the degree of ploidy/binucleation of cardiomyocytes. We then demonstrated that the cardiac-specific deletion of miR205 from the neonatal heart resulted in a 35% increase in the heart mass of 3-month-old mice that was preceded by aberrant cardiomyocyte cell number and cell cycle/Hippo pathway activity in neonatal hearts. In contrast, overexpression of miR205 in the neonatal heart had no impact on heart size or function, most likely reflective of the limited alteration in cell cycle and Hippo pathway kinetics. Collectively, these data support the hypothesis that miR205 plays a critical role in the neonatal cardiomyocytes transition from a Hippo pathway-mediated hyperplastic growth to a hypertrophic growth program (Figure 5).

Our observation establishes that miR205 acts as a potent regulator of perinatal cardiomyocyte cell division, presumably by targeting and modulating the expression of essential cell cycle regulatory factors. Of note, complete genetic deletion of miR205 in mice resulted in neonatal lethality by postnatal day 10 [22]. miR205 directly targets multiple negative regulators of the PI(3)K/Akt pathway, thus promoting the PI(3)K/Akt growth pathway. Furthermore, miR205 plays an essential role in the expansion of skin stem cells and tumours through the modulation of the PI(3)K/Akt pathway [22,38,39,40]. Interestingly, the PI(3)K/Akt pathway is central to the regulation of physiological hypertrophy in the heart [41,42,43]. However, what is most interesting is the observation that miR205 directly targets YAP1 [44] a key component of the Hippo pathway and a mediator of hyperplasia.

The Hippo pathway is a conserved signaling cascade that is essential for the proper regulation of organ growth in Drosophila and vertebrates. In the adult heart, activation of the upstream kinase components of the Hippo pathway prevents adult cardiomyocyte proliferation and regeneration [30], whereas direct activation of Hippo pathway target, the transcription factor YAP1, results in cardiomyocyte proliferation [31,32,33,34,35,36]. This return to proliferation competence with Hippo kinase repression has a profound effect on the adult heart, which regains the capacity to repair and replace cardiac muscle that is lost following infarct damage [32,35,45,46]. Based on these observations, we chose to examine whether there may be a functional intersection in the early perinatal heart between miR205 and the Hippo pathway. Yap, a direct target of miR205 [44], was dramatically upregulated in miR205 null hearts (Figure 2) with a corresponding increase in the expression and activity of CDK1 (Figure 2), a direct regulator of Yap [47,48,49], therefore supporting the concept that the temporal increase in mir205 during the first 7 days post-birth leads to targeted suppression of Yap-mediated cardiomyocyte proliferation and at the same time promoting the PI(3)K/Akt mediated hypertrophic pathway in cardiomyocytes.

However, a limitation of the current model must be acknowledged. Several published reports have demonstrated that excessive Cre recombinase synthesis can have unintended implications in a variety of tissues, including the heart [26,27,28,29,50]. Over-expression of Cre recombinase in cardiomyocytes and cardiac tissue of mice is toxic and induces a variety of cellular stress response pathways and cell death [26,27,28,29], rather than direct induction of hypertrophy. In αMHC-Cre transgenic mice, the heart typically begins to develop dilated cardiomyopathy as early as 3 to 4 months, which ultimately leads to premature death by 1 year of age [27,28,29]. A recent study reported that αMHC-Cre transgenic mice developed cardiac abnormalities, including tumour-like growths in the atrium and substantial fibrosis. These changes were found to be linked to altered protein expression, calcium management issues, and the activation of the ferroptosis signaling pathway [29]. Furthermore, the study concluded that αMHC-Cre animal models are effective in young mice but are not suitable for older animals [29]. However, the specific outcome of Cre recombinase expression causing an increase in heart size (cardiac hypertrophy) has not been widely reported, supporting the notion that the observed increase in cardiomyocyte number and heart mass in the neonatal miR205null hearts is likely due to the blunting of miR205 and not an effect of Cre recombinase.

In contrast to the miR205 deletion model, the overexpression of miR205 did not significantly impact heart size or function. This may be due, in part, to the limited number of miR205 binding sites available in the heart at that time and the fact that additional miR205 has minimal effect, especially since miRNA–mRNA complexes are typically degraded. Furthermore, miR205 targets are transient, since prolonged miR205 expression had an observable effect on heart size. This unexpected finding indicates a more complex regulatory mechanism that is likely redundant in nature and highlights the nuanced role that miRNAs likely play in cardiac physiology, acting not in isolation but as part of a broader, dynamic regulatory system.

The potential therapeutic applications stemming from this study are significant. miR205 appears to play a pivotal role in cardiomyocyte division and the possibility of manipulating it to control the re-entry of these cells into the cell cycle presents an intriguing avenue for myocardial repair. Previous studies have demonstrated the neonatal heart retains a regenerative capacity during the first week after birth [6], which is subsequently lost during maturation. miR205 is transiently expressed during the first week post-birth, and its decrease in expression correlates with the cell cycle exit of the neonatal cardiomyocyte population. As such, it is not unreasonable to suggest that induction of miR205 expression is the nodal point by which neonatal cardiomyocytes initiate the adult cardiomyocyte phenotype. Given our observations, it is reasonable to conjecture that manipulation of miR205 in the adult heart may provide a more tractable proxy to control re-entry of cardiomyocytes into the cell cycle and induce effective repair of damaged myocardium. Several different methods for the therapeutic delivery of miRNAs to the heart have been reported, such as adeno-associated virus mediated delivery, which efficiently transduces cardiomyocytes after either intracardiac or systemic delivery [51,52], and most recently a study has reported the efficacy of a single-dose intracardiac injection of synthetic miRNA that mimics lipid formulation [53].

Indeed, the literature is supported by the core hypothesis that cardiomyocyte cell cycle re-entry can achieve effective myocardial repair. For example, forced expression of a constitutively active form of Yap in the adult heart stimulated cardiac regeneration and improves contractility after MI [35], while Hippo kinase pathway (Mst1, Lats2 and Mob1b) deficiency reversed systolic heart failure after MI [46]. Conversely, AAV9-mediated expression and transgenic expression of constitutively active YAP both resulted in significantly enhanced cardiomyocyte proliferation at the border zone, preserving heart function after MI [35,45].

At this juncture, it is critical to note that miR205 manipulation of the Hippo pathway via Yap expression may offer a more rigorous and palatable method to induce postnatal cardiomyocyte proliferation compared to targeting other elements in the signal cascade. In support of this contention, we and others have noted that Mst-1 can target numerous secondary interactions independent of the Hippo pathway, suggesting that miRNA targeting of Hippo kinases would result in unwanted pleiotropic effects [54,55,56,57].

In conclusion, this study marks a significant step forward in our understanding of neonatal heart development and disease. It positions miR205 as a key regulator in the transition of cardiomyocytes from hyperplastic to hypertrophic growth. The unique expression pattern of miR205 during early heart development and its consequential impact on heart mass and cardiomyocyte density highlight its importance in cardiac biology. While the findings open up potential therapeutic strategies for heart diseases, they also pave the way for further research, especially concerning the long-term effects of miR205 modulation and its interaction with other regulatory mechanisms in cardiac physiology. The study thus serves as a foundation for future investigations into the complex molecular orchestration of cardiac growth and disease.

## 4. Methods and Materials

### 4.1. Microarray Analysis of Micro-RNA Expression

The Genetic Analysis Facility at the Hospital for Sick Children conducted duplicate profiling for each timepoint. The Il-lumina microarray platform comprises a total of 656 miRNA probe sets, excluding Solexa miRNAs, which are based on the Sanger program (Version 12.0). The panels encompass roughly 97% of the miRNAs documented in the miRBase database at the time of the experiment. For the miRNA microarrays, a total of 42 hearts were studied over 14 chips. Three hearts from each timepoint of interest (E19, 1, 3, 5, 7, 10, and 35 days post-birth) were pooled, and processed into experimental duplicates.

### 4.2. Mouse Models

The University of Ottawa Animal Care and Veterinary Services provided housing and treatment for the mice. The mice utilized in our research were accommodated and attended to in accordance with the guidelines set out by the Canadian Council on Animal Care (CCAC) and the protocols established by the University of Ottawa Animal Care Committee protocol #2312 for the present study. As per CCAC guidelines, all adult animals were euthanized in a CO_2_ chamber, followed by cervical dislocation. Neonatal pups were euthanized by decapitation.

#### 4.2.1. Wild-Type Mice

Microarrays and preliminary analysis of miR205 expression and localization were carried out using a colony derived from wild-type 129SV mice purchased from The Jackson Laboratory (Stock #002448).

#### 4.2.2. Cardiac-Specific Deletion of miR205: miR205^fl/fl^ αMHC-Cre^+^ = miR205^−/−^

miR205^fl/fl^ mice were generated from mice from a BL6 background [22,58]. αMHC-Cre mice were purchased from The Jackson Laboratory (Stock #011038). The cardiac-specific miR205 knockout strain was generated by crossing our miR205^fl/fl^ mice with αMHC-Cre^+^ mice to generate mice hemizygous for the miR205 flox locus, with 50% containing the αMHC promoter locus. Mice that had only one copy of the miR205 gene and did not have the αMHCcre gene were bred with other mice who also had only one copy of the miR205 gene but did have the αMHC-Cre gene. The crossings resulted in offspring of which 25% were homozygous for the miR205 floxed allele, and 50% of all offspring had the αMHC-Cre locus. This led to 25% of the mice being miR205^fl/fl^:αMHC-Cre+ (referred to as miR205^−^/^−^). The genotyping of weanlings revealed that the removal of miR205 specifically in the heart did not result in death, and the offspring were capable of surviving into infancy. Subsequently, we successfully bred miR205^fl/fl^ mice with miR205^fl/fl^:αMHC-Cre+ mice, resulting in offspring with selective deletion of miR205 in postnatal cardiomyocytes. Experiments utilized littermate miR205^fl/fl^ pups as controls to compare with the wild-type littermates.

#### 4.2.3. Cardiac-Specific Overexpression of miR205: αMHCrtTA/miR205tetO/DOX^+^ = miR-205^OE^

In order to produce a precise and controllable increase in the expression of miR205 in the heart, two separate lines of mice were created and then bred together. The initial strain is a genetically modified mouse that carries only one copy of a reverse tetracycline transactivator (rtTA) domain located after an αMHC promoter. This led to a mouse that consistently expressed rtTA in any tissue that expressed αMHC, specifically cardiomyocytes. The rtTA protein selectively bound to a particular promoter sequence found in the second strain of mice (tetO). Nevertheless, the functionality of this rtTA protein was dependent on coactivation by doxycycline. This implies that the protein was unable to activate its promoter, regardless of its presence, unless doxycycline was also present.

The second strain contained a hemizygous genetic insertion of miR205 downstream of a tet-operon [58]. This operon contained a specific promoter targeted and activated by rtTA. When bound to the promoter via coactivation by doxycycline, rtTA forced the transcription of miR205.

The crossing of these two strains resulted in progeny, with 1⁄4 inheriting the rtTA allele and 1/4 inheriting the miR205tet allele. In the presence of doxycycline, mouse cardiomyocytes with both alleles would continuously produce miR205. For the remainder of this document, the cardiomyocyte-specific inducible overexpressor of miR205 in the presence of doxycycline is labeled miR205^OE^. The expression of miR205 in the heart was induced by providing the pregnant mice with DOX water at a dosage of 2 mg/mL, two days prior the expected birth of their pups.

### 4.3. DNA/RNA Isolation

Genomic DNA was extracted using Sigma RED Extract-n-AMP PCR kit (Kawasaki City, Japan), as previously described [59,60]. Ear notches were incubated at 55 °C for 15 min in 40 μL extraction solution (Cat. #E7526) and 10 μL tissue preparation solution (Cat. #T3073). Tubes were incubated at 95 °C for 10 min. After chilling for 1 min, each tube received 40 μL of neutralization buffer (Cat. #N3910). The manufacturer’s protocol (Cat. #R4775) was followed for follow-up PCR with primers targeting miR205 flanking regions or additional genomic transgene targets.

TRIzol (Thermo Fisher Cat. #15596026, Waltham, MA, USA) was used to isolate RNA. All stages were carried out at room temperature unless stated. Pooled heart samples were digested in 1ML TRIzol reagent and incubated for 5 min. Each tube received 200 μL of chloroform (Sigma Cat. #288306) for 3 min, then was centrifuged for 15 min at 12,000× *g* RCF at 4 °C in centrifuge tubes. RNA-containing upper aqueous phase was moved to a fresh tube. Each sample received 500 μL of isopropanol (Fisher Cat. #26181) for 10 min. Tubes were spun at 12,000 RCF for 10 min at 4 °C. The particle was resuspended in 1 mL 75% ethanol after discarding the supernatant. Supernatant was eliminated from tubes spun at 7500 RCF for 5 min at 4 °C. Pellets were air-dried for 5 min and resuspended in 50 μL RNAse-free H20. Incubation at 55 °C for 10 min followed. A NanoDrop spectrophotometer evaluated RNA content, and A260/280 determined purity (1.9 or greater was sufficient for follow-up tests).

### 4.4. Western Blot Analyses

Tissue lysate generation and Western blot protocols were followed as previously described [61,62]. Mouse hearts and tail samples were genotyped at 1D, 3D, 5D, 7D, 10D, and 14D. A Polytron Homogenizer with a 7 mm generator homogenized hearts from mice with the same genotype (*n* = 3) in heart lysis buffer (50 mM Tris-HCl (pH 8.0), 200 mM NaCl, 20 mM NaF, 20 mM β-glycerophosphate, 0.5% Nonidet P-40, 0.1 mM Na_3_VO_4_, 1 mM dithiothreitol, 1× protease inhibitor mixture (1 tablet/7.0 mL; Roche, Basel, Switzerland) and phosphatase inhibitor cocktail (0.1 mL/7.0 mL; Sigma)). After 10 min at 4 °C, lysates were centrifuged at 12,000× *g* for 15 min. Supernatant was aliquoted into fresh tubes and kept at −80 °C until SDS-PAGE sample preparation. Bradford protein assay measured and adjusted SDS-PAGE protein content to 1.5 μg/μL. Loading samples were boiled for 5 min with 1× loading dye (Cell Signaling Cat. #56036, Danvers, MA, USA) and 1× DTT (Cat. #14265). Stored samples were −80 °C. Each Western blot utilized 30 μg of total protein, which was loaded onto 4–15% gradient SDS-PAGE gels (BioRad Cat. #4561093, Hercules, CA, USA). The gels were then run through a Tris-Glycine-SDS buffer at 150 V for 45 min, or until the lane marker reached the bottom of the gel. The Transblot Turbo (BioRad Cat. #1704150) was used to blot 0.45 μm polyvinylidene difluoride (PVDF) membrane (Thermo Fisher Cat. #88518) according to manufacturer instructions. ‘High MW’ or ‘low MW’ parameters were utilized for large or tiny proteins. After one hour in 5% bovine serum albumin (BSA), membranes were probed overnight with particular antibodies diluted in BSA. After three five-minute washes in tris-buffered saline (TBS) with 0.1% Tween-20 (TBST), membranes were lightly washed. The membrane was treated with 5% BSA-diluted secondary antibody for one hour at room temperature. Finally, three five-minute TBST washes and another rinse were carried out. Electrochemiluminescence using Clarity substrate (BioRad Cat. #1705060) on Chemidoc XRS+ (BioRad Cat. #1708265) imaging hardware showed blotting.

### 4.5. miR205 RT-qPCR

Hearts were obtained from miR205 wild-type, knockout, and overexpressing mice at 1D, 3D, 5D, 7D, and 10D until a minimum of 3 hearts of each genotype had been processed. Immediately following collection, hearts were rapidly frozen in liquid nitrogen and stored at a temperature of −80 °C. The Trizol technique was employed to purify the total RNA, following the instructions provided by the manufacturer. The reverse transcription of miR205 was performed using Thermo Fisher’s Taqman miRNA Assay (Thermo Fisher Cat. #A25576) in a selective manner. For control purposes, the U6 snRNA was also specifically reverse-transcribed using the identical product (Thermo Fisher Cat. #4427975). Following the reverse transcription (RT) process, quantitative polymerase chain reaction (qPCR) was conducted utilizing the RT-qPCR component of the Taqman miRNA Assay kit. The reactions were conducted on either 96- or 384-well plates using a Roche Lightcycler 480 instrument (Roche Cat. #05015278001, Basel, Switzerland), following the instructions provided by the Taqman manufacturer.

### 4.6. Sectioning, Staining, Immunohistochemistry and Immunofluorescence

The hearts were placed in 70% ethanol after 48 h in 10% formalin and three 30-min PBS washes. Four-chamber view heart sections were generated from paraffin-embedded hearts by the University of Ottawa’s histology core, then subsequently stained with H&E and Masson Trichrome.

Immunofluorescent slides were submerged in xylene 3× for 5 min and cleaned in 100% ethanol 2× for 10 min. The process was repeated in 95% ethanol with two 5-min dH20 washes. Slides were submerged in 1 ug/μL wheat germ agglutinin (WGA) (Thermo Fisher Cat. #W6748) for 10 min. Three 5-min dH20 baths cleaned the slides. The slides were washed three times in dH20 after five minutes in 4′,6-diamidino-2-phenylindole (DAPI) solution (Sigma Cat. #10236276001). Coverslips and Dako mounting medium (Agilent Cat. #S3023, Santa Clara, CA, USA) were applied to each segment and left to dry for an hour. The coverslips were held in place with clear nail polish.

Immunohistochemistry followed Cell Signaling SignalStain Boost Detection Reagent protocol. Citrate unmasked (Cell Signaling Cat. #14746) deparaffinized slides in Coplin jars at 90 °C for 10 min. The slides were cleaned 3× in dH20 for 5 min after cooling. The slides were cleaned 3× in dH20 for 5 min after 10 min in 3% hydrogen peroxide. In a humidified atmosphere, a PAP pen (Thermo Fisher Cat. #008899) created a hydrophobic circle around the slides. The sections were immediately treated with 200 μL of Cell Signaling Animal-Free Blocking Solution (Cat. #15019) for 1 h at room temperature. Antibody diluent (Cell Signaling 8112) with primary antibody for the protein of interest (YAP, Ki67, pH3) replaced blocking solution overnight. Appendix A provides antibody catalog numbers and dilutions. The following day, the slides were washed in dH20 3× for 5 min in Coplin jars. Slides were returned to a humidified chamber and incubated at room temperature for 30 min with three drops of SignalStain Boost Detection Reagent (Cell Signaling Cat. #8114). The slides were then rewashed. A 30 μL quantity of DAB chromogen was diluted into 1 mL DAB Chromogen Diluent (Cell Signaling Cat. #8059). Each item received a 10-min DAB Chromogen solution treatment after three washings. Sections were dehydrated in 95%, 100%, and xylene 2× for 10 s. Sigma 44581/DPX mountant sections.

### 4.7. In Situ Hybridization

Hearts from 5-day-old wild-type mice were immediately mounted in optimal cutting temperature (OCT) compound (Agar Scientific Cat. #AGR1180, Holland, OH, USA) and flash-frozen in liquid nitrogen for in situ hybridization. The hearts were preserved at −80 °C until sectioned. Ottawa’s histology core sectioned the hearts. One hour at room temperature defrosted the heart sections. A miR205-specific RNA probe (Qiagen Cat. #YD00616714, Hilden, Germany) was diluted 1:200 in hybridization buffer (1× SSC buffer, 150 mM Sodium Chloride; 15 mM Sodium Citrate [pH 7.0]) to 200 ng/μL. The tubes were quickly vortexed and denatured at 70 °C for 10 min. A 300 μL quantity of probe mixture was applied to tissue sections on slides after the tubes were briefly spun. The slides were covered and incubated overnight at 65 °C in a humidified room. The next day, the coverslips were carefully removed and the slides were placed in a Coplin jar. Two 30-min washes at 65 °C in solution A (2× SSC buffer) were followed by two 30-min washes at room temperature in 1× TBS-T.

The slides were placed back into the humidified chamber and treated with 300 μL of 10% heat-inactivated fetal bovine serum (FBS) (Thermo Fisher Cat. #16000044) at room temperature for one hour. Following the removal of the blocking buffer, 300 μL of anti-Dig Fab fragments (Sigma-Aldrich Cat. #11093274910) in a solution containing 10% heat-inactivated fetal bovine serum (FBS) in 1× Tris-buffered saline with Tween-20 (TBS-T) (composed of 1 μL alkaline phosphatase (AP), 0.9 mL TBST, and 0.1 mL FBS) was introduced. The coverslips were incubated overnight in a humidified room at a temperature of 4 °C.

Slides were returned to Coplin jars and cleaned 5 times for 20 min in 1× TBS-T. After the first wash, the slides spontaneously lost their coverslips. Slides were incubated in NTMT solution (0.1 M NaCl, 0.1 M Tris-HCl [pH 9.5] 0.05 M MgCl_2_, 1% Tween-20) twice for 10 min at room temperature. Slides were returned to the humidified chamber and stained with NBT/BCIP in NTMT at room temperature in the dark for contrast. Two distilled water washes stopped the reaction. After 20 min in 4% paraformaldehyde (PFA), the slides were washed twice in distilled water. The slides were affixed using Dako mounting material (Agilent Cat. #S3023) and a coverslip, then air-dried and secured with nail varnish.

### 4.8. Microscopy

All microscopic images were collected with a Zeiss Axio Observer platform with color and fluorescence camera. The observer was blinded to the genotype of the sample at time of image collection and analysis. For quantification, the identical sites were photographed for each cardiac part. pH3 was observed at 10× and Ki67 and WGA/DAPI at 20×.

FIJI software (https://fiji.sc (accessed on 8 January 2024)) was used for ImageJ v2.9.0 follow-up analysis for the marking of positive nuclei and exported into Excel to quantify Ki67- and pH3-apositive nuclei. For Ki67, 500 μm by 500 μm areas were used to determine the number of Ki67 positive nuclei. For pH3, we used the complete microscope image, which measured 1 cm × 1 cm. WGA-stained cells were marked and counted as Ki67- and pH3-positive cells in a 200 μM by 200 μM area.

### 4.9. Echocardiography

As previously described [59,63,64], a VisualSonics Vevo preclinical echocardiography imaging device and Vevo software 3100 were used for echocardiography. Three-month-old mice were anesthetized with isofluorane. The mice had their chest hair removed with an electric razor followed by Nair cream to remove all hair. Mice reclined in a supine position on a heated cushion while inhaling isoflurane through a nose cone. Conductive Redux electrolyte cream (Thermo Fisher Cat. #PKR66) was applied to four electrodes placed on the heated pad. The mouse’s four limbs were then securely attached to the electrodes using surgical tape. The Aquasonic ultrasound gel (Thermo Fisher Cat. #PKR01) was applied to the chest. The probe, attached to a mount controlled by a crank, was gradually lowered until the ultrasound software identified a rhythmic motion. The probe was then oriented to view the heart apex, aortic valve, and papillary muscle with each heartbeat. This precise positioning meant that all mice had the same cardiac measurements. B-mode imaging measured left ventricular systole and diastole, length, and area. After rotating the ultrasound probe 90 degrees, M-Mode imaging was employed to measure ventricular diameter during systole and diastole. This mode measured ventricular diameter better. These measurements calculated LV volume, stroke volume, ejection fraction, and left ventricular mass.

### 4.10. Neonatal Mouse Cardiomyocytes Isolation

For anesthesia, neonatal mice received Avertin (0.4 mg/g) intraperitoneally [21]. The heart was revealed by incising the thoracic cavity. A syringe pump delivered 37 °C pre-warmed perfusion fluid. The solution contained 126 mM NaCl, 4.4 mM KCl, 1 mM MgCl_2_, 4 mM NaHCO_3_, 30 mM 2,3-butanedione monoxime, 10 mM HEPES, 11 mM glucose, 0.5 mM EDTA, 0.09% Collagenase Type 2, 0.125% trypsin, and 25 μM CaCl_2_, at pH 7. The solution was pumped through a 26G needle at 4 mL/min. A right atrial incision was created after the needle was inserted into the left ventricle (LV) through the heart’s apex. The mouse was perfused for 5 min. The heart was carefully extracted and placed in a stop solution (126 mM NaCl, 4.4 mM KCl, 1 mM MgCl_2_, 4 mM NaH-CO_3_, 30 mM 2,3-butanedione monoxime, 10 mM HEPES, 11 mM glucose, 0.5 mM EDTA, 100 μM CaCl_2_, 2% BSA, pH 7.4) at 37 °C for 10 min. We sliced the heart into tiny pieces and pulverized it 20–30 times with a wide-bore pipette on a Petri dish. A 100 μm nylon mesh filter was used to remove large particles. The ultimate formalin concentration was 10%, made by adding a 20% solution in equal portions. The cells were fixed for 15 min. The cardiomyocytes were pelletized by centrifuging the fluid at 50 times gravity for 5 min. After the supernatant was discarded, the cardiomyocytes were resuspended in PBS for fluorescent cell labeling.

### 4.11. Statistical Analysis

The miRNA array data underwent analysis via Affymetrix ArrayStar, DAVID, and ingenuity pathway analysis. The micro-RNA data underwent a Log2 transformation and 1-way ANOVA. The normalization of the micro-RNA expression arrays varied due to their disparity in microarray platforms. The data was analyzed by comparing each timepoint with the subsequent one (e.g., 1 day vs. E19). The statistical results underwent correction for multiple testing effects using the false discovery rate (FDR) technique (FDR ≤ 0.05). Tukey’s bi-weight function was utilized for post-hoc analysis. In order to ascertain the count of micro-RNAs that exhibited substantial differential expression, the lists were subjected to a filtering process with a *p*-value threshold of ≤0.05 and subsequently arranged in order of fold-change.

All data comparing experimental group vs. control group are expressed as mean and SEM. Significance of group comparisons was completed by Student’s *t*-test (*p* < 0.05), followed by post hoc statistical analysis (GraphPad Prism 7).

## Figures and Tables

**Figure 1 ijms-25-02206-f001:**
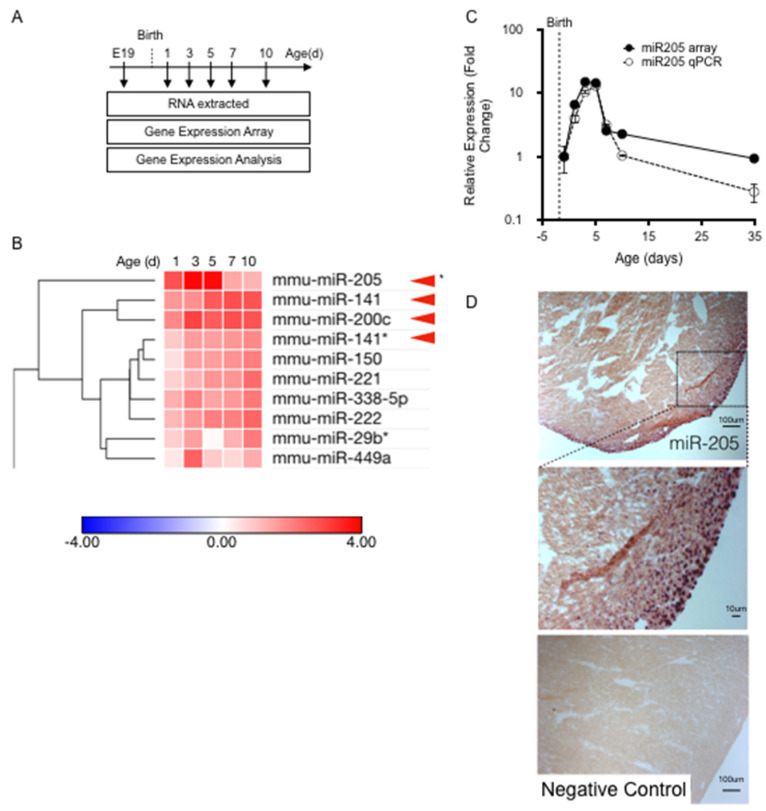
Identification of miR205. (**A**) Total mRNA extracted for temporal miRNA expression analysis. (**B**) Heatmap analysis of temporal miRNA expression. (**C**) RT-qPCR was performed to confirm the miR205 trend observed in the microarray. Mean and SEM, *n* = 3 per time point. (**D**) In situ hybridization showing miR205 localization to the epicardium in 5 day old mouse hearts.

**Figure 2 ijms-25-02206-f002:**
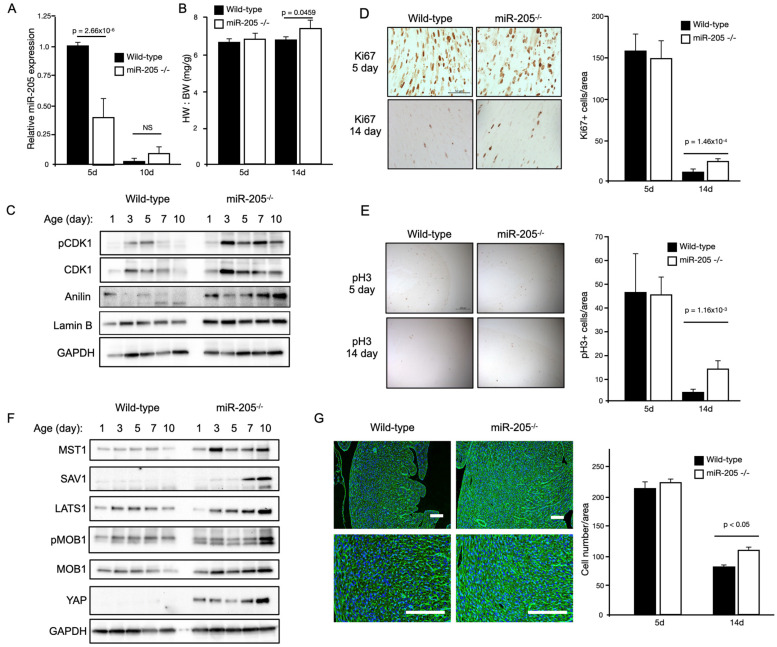
Conditional loss of miR205 in the neonatal heart results in an increased number of cardiomyocytes due to delayed cell cycle exit. (**A**) Significant reduction of miR205 expression (RT-qPCR) in 5-day-old miR205 null hearts. (**B**) Increased heart weight to body weight (HW:BW) of 14-day-old miR205 null hearts. (**C**) Western blot analysis and (**D**,**E**) immunohistochemical analysis of cell cycle markers reveal delayed cell cycle exit in miR205 null hearts. Mean and SEM, *n* = 3 per group per timepoint. (**F**) Persistent Hippo pathway activity (Western blot analysis) in miR205 null hearts. (**G**) Increased cardiomyocyte density of 14-day-old miR205 null hearts. Cellular boundaries were marked by WGA staining (green), and nuclei were labelled with DAPI (blue). Bar = 100 μm. Mean and SEM, *n* = 3 per group per timepoint. For all *p*-values, significance was tested using Student’s *t*-test.

**Figure 3 ijms-25-02206-f003:**
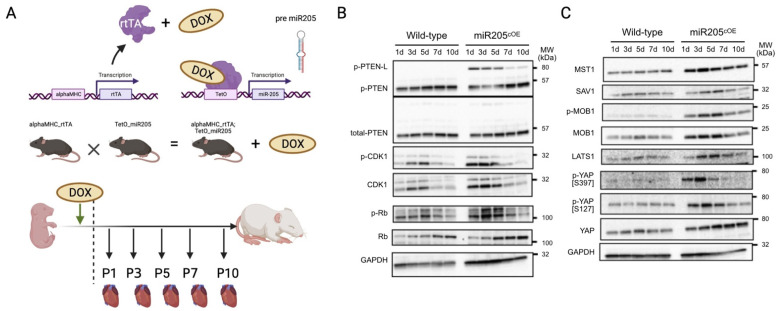
Cardiac overexpression of miR205 leads to dysregulated Hippo pathway in neonatal hearts. (**A**) Generation of a doxycycline inducible cardiac-specific miR205 expression mouse model. Dotted line represents time of birth. Created by BioRender.com (accessed on 12 July 2023) Western blot analysis of (**B**) PTEN and cell cycle markers and (**C**) Hippo pathway.

**Figure 4 ijms-25-02206-f004:**
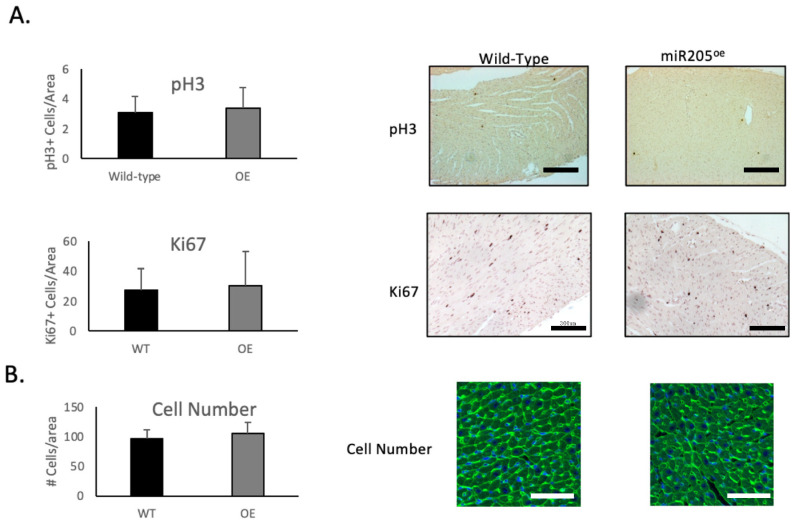
Ectopic expression of miR205 in neonatal hearts has no impact on cardiomyocyte number. (**A**) Immunochemical analysis of cell cycle markers phospho-Histone 3 (pH3) and Ki67 in 14-day-old hearts. Bar = 300 μm. (**B**) Cell density of 14-day-old hearts. Cellular boundaries were marked by WGA staining (green), and nuclei were labelled with DAPI (blue). Bar = 100 μm. Mean and SEM, *n* = 3 per group per timepoint.

**Figure 5 ijms-25-02206-f005:**
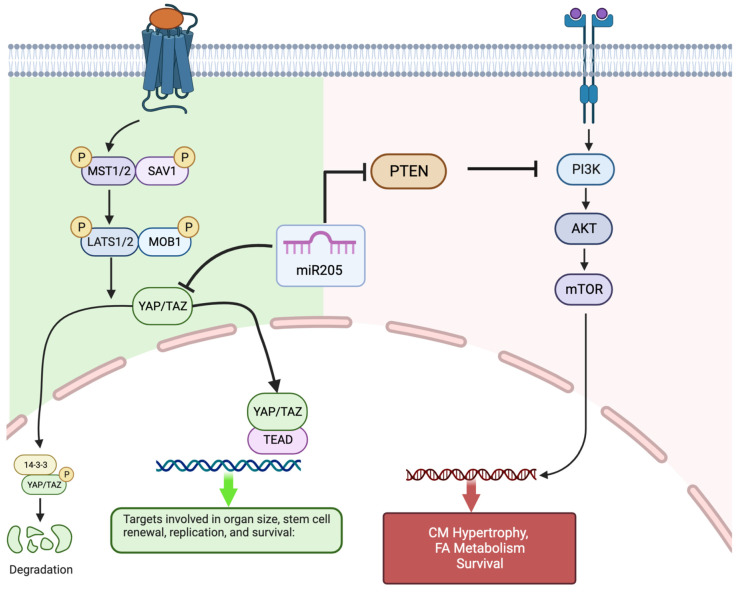
Proposed model of miR205′s relationship to the Hippo (hyperplastic growth) pathway and IP3K/Akt/mTOR (hypertrophic growth) pathway in neonatal cardiomyocytes. miR205, through its interaction with YAP and PTEN, facilitates the transition from Hippo pathway-mediated cardiomyocyte hyperplasia to the PI3K/Akt/mTOR regulated cardiomyocyte hypertrophy. Created with BioRender.com (accessed on 8 January 2024).

**Table 1 ijms-25-02206-t001:** Echocardiographic analysis of 3-month-old mice.

Measurement	Control ^1^	miR205 cKO ^2^	P ^3^
IVS;d (mm)	0.85 ± 0.17	1.11 ± 0.14	0.025
IVS;s (mm)	1.21 ± 0.25	1.45 ± 0.20	ns
LVID;d (mm)	3.33 ± 0.63	3.39 ± 0.81	ns
LVID;s (mm)	2.43 ± 0.81	2.35 ± 0.90	ns
LVPW;d (mm)	0.87 ± 0.13	1.19 ± 0.35	ns
LVPW;s (mm)	1.11 ± 0.27	1.60 ± 0.39	0.078
EF (%)	54.5 ± 18.4	60.5 ± 18.4	ns
LV mass (mg)	76.9 ± 13.6	120.4 ± 17.0	0.003

^1^ *n* = 3; ^2^ *n* = 7; ^3^ ns = not significant.

## Data Availability

The original contributions presented in the study are included in the article/Appendix A, further inquiries can be directed to the corresponding author.

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
