# Peer review of "MicroRNA205: A Key Regulator of Cardiomyocyte Transition from Proliferative to Hypertrophic Growth in the Neonatal Heart"

_ijms, 2024, doi:10.3390/ijms25042206_

Round 1

Reviewer 1 Report

Comments and Suggestions for Authors

Comments on the Quality of English Language

Although the majority of the authors are form Canada and from the US, the language editing requires improvements mainly in terms of correction of several senseless sentences (see for details in the uploaded document)

Reviewer 2 Report

Comments and Suggestions for Authors

Authors address the function of mir205 in the early neonatal period of mouse heart maturation. They use a previously generated conditional allele combined with aMHC-Cre and report knockout consequences in higher cell cycle and heart enlargement. Combined with expression profile, their model is that mir205 is a negative regulator of CM proliferation after birth. They also performed a doxycycline inducible model for overexpression although this did not have obvious impact. There are positive elements in this study but there are also significant issues. I think at minimum the important issues in this study need to be prominently presented in the text.

Significant comments

1. Line 292, the methods description indicates that control mice for the knockout study were fl/fl without Cre. This is simply not acceptable. Cre expression alone has well documented consequences in CMs (e.g., PMID: 16762803 was an early and extreme report but there have been numerous additional studies since then), and including in cell cycle activity (PMID: 33106234). The proper controls are Cre+ littermates. I am reluctant to have the authors repeat all of their experiments, but this is a fundamental flaw in the experimental design - all of the observations might be related to Cre expression rather than mir205 function. My recommendation is to raise this issue very prominently and bluntly in the text (at minimum, certainly at line 123 and in the Discussion), and where possible, for the authors to add additional data using Cre+ controls.

2. A deficiency in the study is that proliferation was not directly measured in mir205 mutants. Cell cycle markers including those shown in Fig 2 are induced as part of polyploidy just as they are in proliferation. The smaller CM size implied by Fig 2G is consistent with more proliferation but is not a proper demonstration of proliferation. It isn’t hard to make this demonstration either, many labs have done such using BrdU followed by analysis in single cell preparations. If no further experimental work is done to show this, then at minimum the statements about proliferation need to be modified.

3. Another deficiency in the study is that no functional demonstration is made to connect Hippo/YAP or any of the other molecular features observed in knockout hearts to the ultimate phenotype at the cellular or whole heart level. The abstract (line 23) is artfully written but somewhat misleading, it isn’t certain that aberrant cell cycle and Hippo activation result in increased CM number and heart mass. This line should be rewritten and the limitation of extrapolating from these observation should be made more clear in the Discussion.

4. Line 164 and beyond, the text and the methods give no indication of when doxycycline was administered. Fig 3 implies that it was started prior to birth (perhaps by provision to near term females?) If so, given the authors’ model that mir205 blocks proliferation, the absence of phenotype in early neonates is puzzling (as genetic manipulations that block neonatal proliferation caused early neonatal phenotypes, one example is PMID: 31597755 and there are at least several others). The authors’ explanation (line 223) that mir205 overexpression induces compensatory upregulation of cell cycle activity doesn’t make sense if higher expression of mir205 is what is suggested to cause the neonatal cessation of proliferation in the first place.

Minor comments to improve text:

Line 45, I think there have been a large number of studies that have addressed mechanisms, genetic and molecular, that are involved in this process.

Line 52, I don’t think that CMs are locked into a nonproliferative state, it seems pretty clear that adult injury reactivates this process. I’d use a different phrasing here.

Line 55, the text claims a change in nuclear morphology. The preceding text only discussed increased ploidy, but to my awareness, the morphology of maturing CM nuclei is not obviously different (other than in some cases being larger for tetraploid nuclei). If authors want to keep this claim, provide citations.

Line 87, text cites authors’ previous paper (ref 20) for this technique but the methods section does not provide any information at all. Include at least a brief description of the method instead of forcing readers to look this up.

Line 102, there are actually several reports related to miR205 in heart (PMID: 34306321, 34078833, 32880505 are just the 3 most recent ones I found).

Line 110, to me it appears that mir205 is abundant in epicardium and subepicardial myocardium, not only epicardium. Also, it appears to be nuclear, which might be something worth mentioning, especially since it is cytoplasmic in at least one prior publication (Fig 2 of ref 20, the source of the knockout allele).

Line 141, the staining used for Fig 2G isn’t stated (presumably WGA, but this needs to be said).

Line 156 and related to Fig 2G, and similarly for Fig 4B, the analysis described here requires properly matched sections of the same angle and portion of the heart. The methods section is completely lacking any description of how this was properly standardized. If sections were prepared by a histology core rather than by the authors themselves, I don’t know if most cores would take appropriate care to do this properly. Including a low power view of the entire heart (if it is a 4 chamber view) would likely address this concern.

Line 279 and lower, authors say aMHC when they mean aMHC-Cre.

Lines 365-366, language is garbled.

Round 2

Reviewer 2 Report

Comments and Suggestions for Authors

Authors have addressed all of my concerns. I hope this has improved the manuscript.